# Leisure Time Use and Adolescent Mental Well-Being: Insights from the COVID-19 Czech Spring Lockdown

**DOI:** 10.3390/ijerph182312812

**Published:** 2021-12-05

**Authors:** Alina Cosma, Jan Pavelka, Petr Badura

**Affiliations:** 1Sts Cyril and Methodius Faculty of Theology, Olomouc University Social Health Institute, Palacký University Olomouc, 779 00 Olomouc, Czech Republic; alina.cosma@hbsc.org; 2Department of Recreation and Leisure Studies, Faculty of Physical Culture, Palacký University Olomouc, 771 11 Olomouc, Czech Republic; jan.pavelka@upol.cz

**Keywords:** leisure, COVID-19, mental health, physical activity, adolescents, free time, active leisure

## Abstract

Background: As leisure—one of the crucial life domains—was completely disrupted by the COVID-19 pandemic, our study aimed to investigate how adolescents spent their leisure time during the Spring 2020 lockdown. Secondly, we aimed to investigate the associations between the perceived changes in leisure time use, the leisure activities adolescents engaged in, and the associations with well-being during the Spring 2020 lockdown in Czechia. Methods: Data from 3438 participants were included in this study (54.2% girls; mean age = 13.45, SD = 1.62). First, the initial number of items measuring leisure, electronic media use, and sports was reduced through Principal Component Analysis (PCA). Multivariate linear regression models tested the associations between leisure domains and mental well-being Results: The amount of leisure time, together with socially active leisure and sports and physical activity, formed the strongest positive predictors of mental well-being, whereas idle activities and time spent on electronic media acted as negative predictors. The amount of time spent doing schoolwork was unrelated to mental well-being. Conclusions: Overall, our results support the idea that leisure as a promoting factor for well-being is not just a matter of its amount but rather of engagement in meaningful and fulfilling activities.

## 1. Introduction

The state of the international public health emergency resulting from the global pandemic declared by the World Health Organization on 11 March 2020 [1] and the subsequent lockdowns imposed by governments across the world had a massive impact on everyday social life and changed the lives of all families [2]. The national school closures in Czechia in March 2020 as part of the government’s response to the COVID-19 pandemic imposed greater demands on school-aged children and adolescents, as well as their family members, with regard to structuring time and fulfilling all their schoolwork tasks [3]. Remote learning was challenging for the adolescents but also for the adults involved in organizing this process as it required active collaboration between parents, teachers, schools, and school administrators in the learning process over several months [4]. Besides the distorted school routine, the content of out-of-school hours was also anything but normal. Organized sport activities were prohibited both indoors and outdoors [5], the opportunities for peer face-to-face interactions were limited, and all-day residence under the care of parents resulted in changes to behaviors [6]. Overall, for adolescents, this meant that their daily routine, in which school and organized leisure activities used to take up a significant portion of their time, was gone. Thus, from a public health perspective, it is important to understand how adolescents structured their free time during the first COVID-19 lockdown, and how school and different types of leisure activities related to their well-being.

It is widely recognized that leisure activities provide adolescents with unique developmental opportunities for socialization and learning as well as increased well-being and good mental health [7,8]. These findings have been supported in organized settings [7,9] but also in unstructured [10,11] or self-organized [12] activities usually carried out in the companionship of peers. Psychological mechanisms such as detachment-recovery, autonomy, mastery, meaning, and affiliation have been positioned to explain how leisure activities promote mental well-being [13]. However, the societal changes prompted by the COVID-19 pandemic forced adolescents to seek other ways of spending their time, focusing on activities they could do solely inside their homes. These included painting, playing board games, reading, or playing a musical instrument [14], but a lot of time was, expectably, devoted to screen-based activities too [15,16]. Given that each of these leisure activities has its own unique characteristics, we would expect different associations between each leisure domain and mental well-being. More specifically, in line with previous theoretical conceptualizations outlined [13], leisure activities that satisfy multiple psychological needs (such as meeting with friends or exercising) would be expected to promote a higher sense of mental well-being compared to activities that fulfil a single psychological need (such as idle leisure activities).

Relying on electronic devices for schoolwork, social interactions, and entertainment has become the new norm for adolescents during the COVID-19 pandemic [17], with increases in recreational screen time being reported across various cultural contexts [18]. At a broader level, the COVID-19-related school closure for several months led to an increase in physical inactivity and sedentary behavior [4,16]. Apart from the risk of an increase in the prevalence rates of overweight and obesity, this was also detrimental to the mental health and well-being of adolescents [16]. Recent review evidence points to the fact that total screen time has small to very small effects on subsequent depressive symptoms [19]. However, in the midst of the COVID-19 pandemic, there have been reports that recreational screen time was negatively associated with mood after controlling for the relevant variables (i.e., physical activity and body mass index) [20]. Thus, we would expect a negative association between screen time and mental well-being among Czech adolescents during the 2020 Spring lockdown.

Previous research suggested that it is necessary to focus on the quality and content of leisure time, which can provide a way for young people to connect positively with their peers and communities, and to promote constructive meaning-making in their lives [21,22]. As leisure was one of the life domains that was completely disrupted by the COVID-19 pandemic, the main aim of the present study was to investigate how adolescents spent their free time (i.e., leisure time) during the Spring 2020 lockdown. Secondly, we aimed to investigate the associations between the perceived changes in leisure time use, the leisure activities the adolescents engaged in, and the associations with well-being during the Spring 2020 lockdown in Czechia. More specifically, we addressed the following research questions:

*RQ1:* How did Czech adolescents spend their time during the Spring 2020 COVID-19 lockdown?

*RQ2:* What was the association between adolescent leisure and school time use (as well as perceived changes in the use of time compared to the pre-COVID era) with mental well-being?

*RQ3:* What were the associations between different leisure domains and adolescent mental well-being during the Spring 2020 COVID-19 lockdown?

## 2. Materials and Methods

### 2.1. Sample and Procedure

The present study built upon the methodology of Health Behaviour in School-aged Children (HBSC), a WHO collaborative trans-national study [23]. In May 2020, 234 randomly selected schools (International Standard Classification of Education levels 1 and 2) across all 14 Czech administrative regions were invited to join an online survey investigating the adolescents’ use of time and health behaviors during the Spring 2020 school closures. In each school that was willing to take part in the survey (*n* = 144), adolescents enrolled in the fifth, seventh, and ninth grades (corresponding to the age categories of 11, 13, and 15 years, respectively) were invited to take part in the survey. Data collection took place in June 2020 via web-links sent to the respondents by their class teachers, with the exception of the data from four schools, which preferred a pen-and-paper survey. After receiving the questionnaire link from their class teachers, the pupils were asked to fill it in at home. In total, we collected 3384 online questionnaires and 232 pen-and-paper questionnaires. Next, the data file was checked and 178 respondents were excluded from the analyses because of reporting age out of the valid age range for the given grade, contradictory responses or excessive number of missing values throughout the questionnaire, etc. The sociodemographic characteristics of the final sample are presented in Table 1. The response rate at the individual level was 19%. However, it varied across schools from 1% up to 72%. Participation in the study was anonymous and voluntary and no incentives were offered for participation. The respondents could withdraw from the survey at any time or skip questions that they were not comfortable with. Active consent was obtained at the school and pupil level, whereas passive consent was used at the parent level.

### 2.2. Instruments

#### 2.2.1. Leisure-Time Activity Participation

*Time spent on leisure* was assessed with a single item developed for the purpose of this study. Adolescents were asked how much free time they had on school days during coronavirus when schools were closed in Spring 2020 to engage in activities they enjoyed. The responses ranged from (1) “none at all” to (9) “about seven or more hours a day”. The responses were recoded to reflect the actual time with scores ranging from (0) “none at all” to (7) “about seven or more hours a day”.

*Time spent doing schoolwork* was assessed with a single item developed for the purpose of this study. The adolescents were asked how much time they spent doing schoolwork during coronavirus when schools were closed in Spring 2020. The responses ranged from (1) “none at all” to (9) “about seven or more hours a day”. The responses were recoded to reflect the actual time with scores ranging from (0) “none at all” to (7) “about seven or more hours a day”.

*Perceived changes in comparison to pre-lockdown.* The adolescents reported whether they spent more or less time on leisure (free time) or schoolwork. The responses ranged from (1) “definitely more” to (5) “definitely less”. For the purpose of this study, for each of the items, two sets of dummy variables were created (i.e., More time spent on leisure; More time spent on schoolwork, and Less time spent on leisure; Less time spent on schoolwork) with the same amount of time spent on leisure/schoolwork during and before the Spring 2020 lockdown set as a reference category.

*Electronic Media Use (EMU)* was measured with four items asking about the time spent (i) watching videos, films, or series; (ii) on social media; (iii) browsing the internet, and (iv) playing video games on PC, console, tablet, etc. The responses ranged from (1) “not at all” to (9) “about seven or more hours a day”. The responses were recoded to reflect the actual time, with scores ranging from (0) “not at all” to (7) “about seven or more hours a day”.

*Leisure.* The adolescents were asked how often during the school closure resulting from COVID-19 that they engaged in (i) reading books; (ii) going out for walks; (iii) playing board games; (iv) playing active video games (e.g., Beat Saber, Ring Fit Adventure, Just Dance, PowerBeatsVR); (v) meeting friends outside; (vi) going outdoors; (vii) playing a musical instrument; (viii) idling; (ix) napping; (x) creative writing; (xi) photography, videos, vlogging. The responses ranged from (1) “daily” to (4) “almost never”. All the responses were recoded in such a way that the highest values (4) indicate daily engagement.

*Sports and physical activity.* The adolescents were asked how often during the school closure because of COVID-19 that they engaged in (i) exercising at home; (ii) exercising outdoors; (iii) running; (iv) riding a bike; (v) skateboarding; (vi) walking or hiking. The responses ranged from (1) “daily” to (4) “almost never”. All the responses were recoded in such a way that the highest values (4) indicate daily engagement.

The items on specific leisure activities used in the study were under development prior to the COVID-19 outbreak as part of the activities (unpublished so far) of the HBSC Leisure Conceptual Group. For the survey, they were reworded so that they reflected the COVID-19 lockdown. The questions regarding perceived changes were created ad hoc for the purpose of the COVID-19 lockdown survey.

#### 2.2.2. Mental Well-Being

Two measures were used to measure adolescent mental well-being. *Life satisfaction* was measured using the Cantril ladder, a visual analogue 11-point scale for rating how adolescents feel about their life at present with responses ranging from (0) “worst possible life” to (10) “best possible life”. This ladder is easily understood and a reliable instrument in adolescent populations [24]. The *WHO-5 Well-being Index* is a well-validated mental well-being measure comprising five items (i.e., I have felt cheerful and in good spirits; I have felt calm and relaxed; I have felt active and vigorous; I woke up feeling fresh and rested; My daily life has been filled with things that interest me) [25]. The adolescents were instructed to rate each statement and indicate which was the closest to how they had been feeling over the last two weeks. The response options ranged from 0 (at no time) to 5 (all of the time).

*Gender and Age.* The adolescents reported the year and month in which they were born, and whether they were a boy or a girl.

### 2.3. Analytical Approach

Our analytical strategy followed two steps. First, we reduced the initial number of items measuring leisure, EMU, and sports through Principal Component Analysis (PCA). As our instruments of interest were measured on different scales, we decided to run three separate PCAs. This exploratory technique was selected instead of confirmatory factor analysis so as not to restrict selection of the leisure items by making theoretical assumptions about their associations and to allow the evaluation of the emergence of a simplified structure of leisure constructs through a data-driven procedure (Vacciano and Bolano, 2020). The PCA was performed with oblique Promax rotation using the IBM SPSS Statistics software (version 25). Applying oblique rotation instead of an orthogonal type (e.g., varimax) allowed us to relax the assumption of having uncorrelated latent traits. A threshold of 0.40 was set for the factor loadings. Internal consistency was assessed by computing the Cronbach’s Alpha.

In the second step, we ran multivariate linear regression models, which tested the associations of the different leisure components, time spent doing leisure and schoolwork, and perceptions of time allocation resulting from the COVID-19 lockdown life satisfaction (Model 1) and WHO-5 Well-being Index (Model 2) through multivariate linear regression models. All linear regression models were controlled for age and gender. In a final step, the aforementioned models were run separately for boys and girls to evaluate possible gender effects.

## 3. Results

Overall, 3438 participants were included in this study (54.2% girls; M_age_ = 13.45, SD = 1.62). On average, the Czech adolescents indicated that they spent 2.8 h (SD = 1.6) per day on schoolwork during the Spring 2020 lockdown, whereas they devoted five hours (SD = 1.9) a day to the leisure activities they enjoyed. The boys reported investing less time in schoolwork (2.6 h) and more time in leisure activities (5.2 h) than the girls (2.9 h for schoolwork and 4.9 for leisure activities; both at *p* < 0.001).

In terms of perceived changes in time use compared to the “pre-COVID” period, 52% of the respondents felt that they spent more time doing schoolwork, 22% about the same, and 26% felt that the amount of time spent on schoolwork had rather decreased. Concerning leisure, 66% considered that they had more leisure time than before the lockdown, 20% reported no change, and 14% indicated that the amount of their leisure time decreased when the schools were closed in Spring 2020. Neither of the perceived change variables differed by gender (*p* = 0.246 and 0.576 for schoolwork and leisure activities, respectively).

### 3.1. Principal Component Analysis

Three separate PCAs were performed. The results are outlined in Table 2.

#### 3.1.1. Electronic Media Use

This component was formed by four different items (i.e., watching videos, social media use, internet browsing, and gaming) with factor loadings ranging from 0.577 to 0.761. All these activities relate to EMU. The Kaiser–Meyer–Olkin (KMO) value of 0.689 and Bartlett’s test (*p* < 0.001) indicated the adequacy of using factor analysis in our sample. All the retained components explained 50.5% of the total variance and had an acceptable reliability (Cronbach’s alpha = 0.66).

#### 3.1.2. Leisure

The PCA analysis revealed three separate components. Overall, these explained 54.1% of the variance and the Kaiser–Meyer–Olkin (KMO) value of 0.689 and Bartlett’s test (*p* < 0.001) indicated the adequacy of using factor analysis in our sample. The *social active leisure* component (19.22% variance, Cronbach’s alpha = 0.60) contained three correlated items (i.e., going out, going outdoors, and meeting friends outside) with factor loadings ranging from 0.648 to 0.769. This component simultaneously links to face-to-face interactions, engagement in social interactions, and proactive use of time. *Cultural creative leisure* (13.04% variance, Cronbach’s alpha = 0.54) contained four correlated items (i.e., reading books, drawing, creative writing, and playing a musical instrument) with factor loadings ranging from 0.536 to 0.695. This component includes solitary but purposeful activities that adolescents could do at any time whilst at home. The third emerging component was *Idle* (11.89% variance, Cronbach’s alpha = 0.31). This component contained four items (i.e., napping, idling, photography, and active video games) and reflected a passive use of free time. This component had the lowest factor loadings (0.401 to 0.704) and internal consistency (Cronbach’s alpha = 0.31). The item on playing board games did not load on any of the emerging components and it was therefore discarded from the analyses.

#### 3.1.3. Sports and Physical Activity

The final PCA confirmed the emergence of one component associated with practicing sports and physical activity. This component (39.04% variance, Cronbach’s alpha = 0.68) was composed of five items with factor loadings ranging from 0.511 to 0.768. Thus, it captured a wide range of exercising activities that adolescents could do either by themselves or together with others.

### 3.2. Correlations between Leisure-Related Components

The time spent doing leisure activities correlated negatively with the time spent doing school-related activities (r = −0.229) and correlated positively with EMU (r = 0.174) (Table 3). The social active leisure component correlated positively with the sports and physical activity components (r = 0.609) and cultural creative leisure (r = 0.188) and correlated negatively with EMU (r = −0.115). On the other hand, the EMU component correlated positively with the idle leisure component (r = 0.428) and correlated negatively with the sports and physical activity (r = −0.155) and cultural creative leisure components (r = −0.191).

### 3.3. Associations with Mental Well-Being

Table 3 outlines the correlations between the main variables included in this study. Across both outcomes, adolescent girls and older adolescents reported poorer mental well-being. The time spent by the adolescents doing leisure activities during the Czech Spring 2020 lockdown correlated positively with the WHO-5 Well-being Index (r = 0.210), life satisfaction (r = 0.157), and EMU (r = 0.174) and associated negatively with cultural creative leisure (r = −0.080) and sports and physical activity (r = −0.080). Importantly, the time spent doing leisure activities correlated negatively with the time spent doing schoolwork (r = −0.229). Moreover, the time spent doing schoolwork did not correlate with any of the mental well-being outcomes. The social active leisure and sports and physical activity components had the strongest associations with the mental well-being outcomes (r = 0.159 and 0.168 with life satisfaction and r = 0.210 and 0.241 with the WHO-5 Well-being Index, respectively).

Multivariate regressions were used to analyze the associations between different leisure components, time spent doing schoolwork during the Spring 2020 lockdown, and mental well-being indicators (i.e., life satisfaction and the WHO-5 Well-being Index). The results are reported in Table 4. The overall model predicting life satisfaction accounted for 12% of the total variance (F = 30.643, *p* < 0.001), with the strongest predictors being time spent doing leisure activities (ß = 0.165, *p* < 0.001) and the sports component (ß = 0.110, *p* < 0.001). The idle component (ß = −0.104, *p* < 0.001) and EMU (ß = −0.081, *p* < 0.001) were negative predictors for life satisfaction. The importance of leisure is further illustrated by the positive associations between those who perceived that they had more leisure time compared to those who said they spent about the same time doing leisure activities (ß = 0.054, *p* < 0.015). In line with that, those who perceived a decrease in their amount reported lower life satisfaction (ß = −0.053, *p* < 0.018). Schoolwork-related variables were not associated with life satisfaction.

Similarly, the time spent doing leisure activities during the Spring 2020 COVID-19 lockdown was the strongest positive predictor of the WHO-5 score (ß = 0.183, *p* < 0.001). The social active leisure (ß = 0.146, *p* < 0.001) and sports and physical activity (ß = 0.158, *p* < 0.001) components were positively associated with WHO-5, whereas the idle component was a negative predictor (ß = −0.158, *p* < 0.001). The schoolwork-related variables and the cultural creative and EMU components were not associated with WHO-5. The overall model explained 20% of the variance in WHO-5 (F = 52.546, *p* < 0.001).

The gender desegrated multivariate analyses (Appendix A) show that the associations of different time allocation during COVID-19 lockdown and, in particular, specific leisure domains and mental well-being are robust and consistent across both genders. Only few gender differences were observed. For both outcomes, the EMC component was a significant negative predictor just for girls (ß = −0.112 for life satisfaction, and ß = −0.087 for WHO-5; *p* < 0.05). Perceived changes in time spent doing leisure seemed to matter more for adolescent girls’ mental well-being, compared with boys. As such, having more (ß = 0.079, *p* = 0.006) leisure than before the COVID-19 lockdown was a significant predictor for the WHO-5 well-being index only for girls. Concerning life satisfaction, the perception of having less leisure than before the COVID-19 lockdown was a significant predictor in adolescent boys (ß = −0.068, *p* = 0.047). In addition, adolescent girls who reported having more leisure than before the COVID-19 lockdown reported higher level of life satisfaction (ß = 0.062, *p* = 0.035).

## 4. Discussion

This paper aimed to examine how Czech adolescents spent their free time during the Spring 2020 COVID-19 lockdown, with a specific focus on the associations between the perceived changes in leisure time use and schoolwork and well-being during the aforementioned period. Given the impact of the imposed measures as part of the Spring 2020 lockdown, more specifically, we also aimed at identifying the domains of leisure activities in which adolescents engaged and how these related to their mental well-being.

Firstly, our findings show that physical activity and sports were the domain of leisure activities among the Czech adolescents during the first COVID-19 lockdown with the highest factor loadings. This domain correlated positively with social active leisure. These findings confirm previous research showcasing that sports are often rated as the most important and interesting leisure activity by adolescents [26] and they are associated with a better social life and more outdoor time [27]. Another important domain concerning how adolescents spent their free time centered around the use of electronic media. Through the measures implemented as part of the Spring 2020 lockdown, adolescents had to rely exclusively on electronic devices to continue school work and keep in touch with friends [17]. Our results also show that this component correlated strongly and positively with the idle leisure component and negatively with the cultural creative and sports and physical activity components. Thus, it could be argued that adolescents sometimes engage in these activities because there is nothing better to do. This agrees with previous research linking it with feelings of boredom, amotivation, or a perceived lack of community activities [28] or even risky sensation-seeking behaviors [29]. However, in the case of smartphone use, as this is central to adolescents’ lives nowadays, recent evidence shows that smartphones might be used to alleviate boredom during unstructured leisure but also offer low-commitment leisure opportunities to relax and interact with friends [30].

Overall, the perceived amount of leisure time was the strongest predictor for adolescent mental well-being. One immediate consequence of the national COVID-19 lockdown and all its accompanying measures, which severely disrupted the lives of adolescents (e.g., online school, sports clubs closed, limited interactions outside the household, etc.), was the increase in the amount of discretionary time [31]. Our results suggest that during the first Czech COVID-19 lockdown, this perceived increase in leisure time was a protective factor for adolescent mental well-being. Previous qualitative research shows that how young people decide to use their leisure time can make the difference between a healthy or unhealthy adolescent [32]. However, recent research emphasizes that there is a critical threshold for how much discretionary time is beneficial for individuals; too much or too little free time has proved to be detrimental [33]. If too little discretionary time can activate feelings of stress, too much discretionary time, on the other hand, can induce feelings of lacking productivity [33].

By contrast, the amount of time spent doing schoolwork was not associated with mental well-being. Previous studies report systematic associations between having too much schoolwork and poor mental well-being [34,35]. However, despite the fact that more than half of the participants reported having spent more time doing schoolwork than before the COVID lockdown, these perceived changes were not associated with mental well-being. During the first COVID-19 lockdown the reported time devoted to schoolwork was actually shorter, compared to the time usually spent at school. On average, Czech 11–15-year-olds would usually spend roughly 4–5 h of “frontal education” at school a day and additional time spent doing homework. Thus, it could be that adolescents might have just perceived “homeschooling” during the lockdown to be more intense. In light of the previous research [33], it is also plausible that schoolwork provided the respondents with feelings of productivity to counterbalance the “unprecedented amount” of time to be spent outside formal structures (in schools or organized activities). The aforementioned lack of associations between schoolwork and mental well-being could also be explained by how Czech adolescents generally report school and schoolwork experiences. Despite a medium ranking in the 2018 PISA results among OECD countries [36], Czech adolescents seem to be less connected to school dimensions (i.e., liking school, teacher and classmate support) than their European counterparts [37].

Most importantly, the amount of leisure time, together with social active leisure and sports and physical activity, acted as the strongest positive predictors of mental well-being. These findings are in line with previous findings showing that active leisure is a strong driver of well-being [7,38]. Previous research also shows that frequent participation in outdoor play activities prior to the pandemic provided lasting resilience against drops in mental well-being during the pandemic [39]. These findings are similar to our results as we could see that those adolescents who reported engaging in less leisure time than before the COVID-19 pandemic also reported lower levels of mental well-being.

It is important to note that all the domains of leisure activities identified here were associated in the same direction and strength with the mental well-being indicators, as active leisure domains (i.e., physical activity and pro-socialization activities) were protective factors for mental well-being, whereas passive leisure domains (i.e., idle) were risk factors. It has been speculated that leisure functions as a therapeutic method involving personally meaningful and intrinsically interesting activity, leading to enhanced relaxation, self-determination, and self-efficacy, as well as distracting people from negative events [40]. In line with recent results [39], our findings suggest that physical activity and pro-socialization activities are the most robust predictors of positive mental well-being. On the other hand, our findings also suggest that engaging in passive leisure activities (such as napping, idling, photography, etc.) is associated with a decrease in mental well-being. However, in the case of adolescents, these would rather be passive leisure activities done just to keep busy and fight boredom. Similar negative effects have been reported elsewhere [38]. Furthermore, despite adolescents spending a significant amount of time on social media or connecting virtually with friends, the EMC component was negatively associated with mental well-being, similarly to findings reported in Canada, e.g., [15]. As these associations were seen only in girls, our findings echo previous findings that the association between moderate and heavy digital media use and low mental well-being is usually larger for girls [41]. Finally, the cultural creative component was not associated with any of the mental well-being outcomes. This is at odds with recent evidence showing that engagement in creative activities during COVID-19 was associated with higher levels of mental well-being [42].

These findings might be interpreted with consideration being given to the context in which these data were collected, namely the first COVID-19 national lockdown in Spring 2020. The 3-month closure of schools manifested in a low response rate at the individual level. This limits the generalizability of the present findings substantially. However, respondents from all the administrative regions were included, ratio of boys vs. girls was close to the entire Czech population of the target age categories, and data indicating socioeconomic status were comparable with the previous HBSC surveys conducted in Czechia. Whilst some of our findings do provide further insights into how adolescents have structured their leisure time and how these different activities were associated with their mental well-being, it is worth mentioning that most of these associations would also probably hold true in a non-COVID-19 situation. The lockdown might have made some of these clusters of activities more salient; however, we would expect that even in a situation in which there were no COVID-19 lockdowns, active leisure activities and a sufficient amount of leisure time would, in general, continue to be strong positive predictors of mental well-being. Notwithstanding, an important limitation of the current investigation is represented by the low composite reliability of the different leisure components. This could be due to the low number of items within each component but also due to the specific context in which data collection was undertaken (i.e., lockdown), which inherently affected the adolescents’ responses to these items. Future research ought to replicate these analyses in a non-pandemic situation. Another further limitation worth mentioning is the cross-sectional nature of our data. Therefore, we cannot imply any directionality nor causality. Lastly, given the circumstances, the instruments focusing on leisure activities during the COVID-19 pandemic could not be properly validated prior to their use in the study. Future studies should focus on testing the longitudinal associations reported here in non-lockdown conditions.

## 5. Conclusions

The national COVID-19 lockdowns presented challenges for adolescents worldwide. As schools were closed and organized activities or any other social and cultural events were strictly prohibited, the first national COVID-19 lockdown in Czechia had an impact on the routines and activities of adolescents. Our findings suggest that in periods such as these, the time spent doing leisure activities, as well as the time spent outdoors meeting with friends or playing and doing sports, plays a critical role in fostering positive mental well-being. Overall, our results support the idea that leisure as a promotive factor for well-being is not just a matter of its amount but rather of engagement in meaningful and fulfilling activities. These could be seen as drivers of resilience in challenging times, and public health policies and national and regional communities should facilitate outdoor recreation opportunities for young people during times of crisis.

## Figures and Tables

**Table 1 ijerph-18-12812-t001:** Sociodemographic characteristics of the sample (*n* = 3438).

		*n*	Percent	Mean (SD)
Gender				
	Boy	1574	45.8	
	Girl	1866	54.2	
Grade				
	5th	1080	31.4	
	7th	1335	38.8	
	9th	1025	29.8	
Residence (no. of inhabitants)				
	>50,000	556	16.5	
	10,000–49,999	997	29.6	
	2000–9999	741	22.0	
	<2000	1068	31.7	
Age				13.45 (1.62)

**Table 2 ijerph-18-12812-t002:** Principal component analyses (rotated loadings).

Item Name	Components				
	1 *	2 *	3	4	5 *
	Electronic Media Use	Social Active Leisure	Cultural Creative Leisure	Idle	Sports
Measure 1					
Videos, films, series	0.761				
Social media	0.759				
Internet browsing	0.729				
Games on PC, console, tablet	0.577				
Measure 2					
Going out		0.769			
Going outdoors		0.746			
Meeting friends outside		0.648			
Reading books			0.695		
Drawing			0.660		
Creative writing			0.634		
Musical instrument			0.536		
Napping				0.704	
Idling				0.586	
Photography				0.429	
Active video games				0.401	
Measure 3					
Exercise outdoors					0.768
Running					0.710
Cycling					0.603
Exercise at home					0.576
Skateboarding; inline skating					0.540
Walking, hiking					0.511
Total variance	50.527	19.219	13.044	11.885	39.039
Composite reliability	0.66	0.60	0.54	0.31	0.68
KMO	0.689			0.689	0.753
Bartlett’s test of significance	0.000			0.000	0.000

* Three separate principal component analyses were run, one for each measure.

**Table 3 ijerph-18-12812-t003:** Correlations between the variables included in the study.

Variable	1	2	3	4	5	6	7	8	9
1. Leisure (h)	—								
2. Schoolwork (h)	−0.229 **	—							
3. Social active leisure	−0.006	−0.006	—						
4. Cultural creative leisure	−0.080 **	0.177 **	0.188 **	—					
5. Idle leisure	0.027	−0.137 **	0.055 **	0.033	—				
6. Sports and physical activity	−0.037 *	0.014	0.609 **	0.261 **	0.048 *	—			
7. Electronic media use	0.174 **	−0.080 **	−0.115 **	−0.191 **	0.428 **	−0.155 **	—		
8. Life satisfaction	0.157 **	0.005	0.159 **	0.012	−0.149 **	0.168 **	−0.131 **	—	
9. WHO-5 Well-being index	0.210 **	−0.035	0.255 **	0.043*	−0.170 **	0.241 **	−0.120 **	0.482 **	—

** *p* > 0.001; * *p* > 0.01.

**Table 4 ijerph-18-12812-t004:** Associations between different leisure components, school, and mental well-being.

	Life Satisfaction					WHO-5 Well-Being Index				
Model	B	SE	ß	t	*p*	B	SE	ß	t	*p*
(Intercept)	8.928	0.329		27.167	<0.001	76.154	4.397		17.321	<0.001
Gender (boys vs. girls)	**−0.350**	**0.069**	**−0.097**	**−5.107**	**<0.001**	**−7.637**	**0.925**	**−0.152**	**−8.252**	**<0.001**
Age	**−0.124**	**0.021**	**−0.111**	**−5.838**	**<0.001**	**−1.546**	**0.285**	**−0.100**	**−5.418**	**<0.001**
Leisure (hrs)	**0.156**	**0.018**	**0.165**	**8.412**	**<0.001**	**2.385**	**0.249**	**0.183**	**9.581**	**<0.001**
Schoolwork (hrs)	0.040	0.023	0.035	1.745	0.081	0.184	0.308	0.012	0.599	0.549
Perceived more leisure ^†^	**0.204**	**0.084**	**0.054**	**2.441**	**0.015**	**3.662**	**1.134**	**0.070**	**3.230**	**0.001**
Perceived less leisure ^†^	**−0.273**	**0.115**	**−0.053**	**−2.375**	**0.018**	**−3.363**	**1.546**	**−0.047**	**−2.175**	**0.030**
Perceived more schoolwork ^†^	−0.013	0.084	−0.004	−0.152	0.879	0.216	1.132	0.004	0.191	0.849
Perceived less schoolwork^†^	−0.081	0.096	−0.020	−0.839	0.402	−0.702	1.293	−0.012	−0.543	0.587
Social active leisure	**0.139**	**0.041**	**0.077**	**3.384**	**<0.001**	**3.679**	**0.558**	**0.146**	**6.590**	**<0.001**
Cultural creative leisure	−0.040	0.036	−0.022	−1.117	0.264	0.179	0.487	0.007	0.367	0.714
Idle leisure	**−0.190**	**0.037**	**−0.104**	**−5.121**	**<0.001**	**−3.988**	**0.502**	**−0.158**	**−7.948**	**<0.001**
Sports and physical activity	**0.201**	**0.042**	**0.110**	**4.772**	**<0.001**	**3.986**	**0.571**	**0.158**	**6.979**	**<0.001**
Electronic media use	**−0.147**	**0.038**	**−0.081**	**−3.863**	**<0.001**	−0.721	0.514	−0.029	−1.402	0.161

^†^ Those who perceived no change in the amount of their leisure time or time spent on schoolwork served as a reference group. Statistically significant (p<0.05) associations are indicated in bold.

## Data Availability

Data are available on reasonable request from the last author of the study (petr.badura@upol.cz).

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
