# Peer review of "Leisure Time Use and Adolescent Mental Well-Being: Insights from the COVID-19 Czech Spring Lockdown"

_ijerph, 2021, doi:10.3390/ijerph182312812_

Round 1
Reviewer 1 Report
The submission examines how initial COVID lockdowns impacted leisure and school time use among Czech adolescents. Specifically, the study examined the association between types of leisure time and school time (as well as perceived changes from pre-COVID times) and well-being as per the Cantril ladder and the WHO-5 Well-being index. Changes in how youth spend their time and the effects of those changes on mental health are important to understand in light of the COVID pandemic and resulting lockdowns, and to better plan for future similar events. As such, the study is essential to the field. The following are offered to strengthen the submission:
- There is some confusion regarding the sample. Line 103 states the web-links were sent to 3,384 respondents plus 232 pen-and-paper (3,615 total). Yet line 105 states the response rate was 19%. Explanation on how those numbers lead to the final sample of 3,438 would be a welcomed addition.
- Further, if the 19% response rate is correct, the authors should include in the discussion section some mention of caution regarding that rate and generalizability of the results. Some indication of how responders compare to all youth across the schools, at least in terms of age and gender would be useful as well.
- In the measures section, both EMU and Leisure state “playing video games” is included as a component. Table 2 helps a little bit as it distinguishes “Games on PC/console/tablet” from “Active video games”. However, a little more detail on how those two types of video games differ would be useful so that is does not appear that the same thing (video games) is included in both concepts.
- Many of the alpha values from the PCA are very low (.6 and below). The authors should mention how this may affect results/findings. I was very surprised to not see mention of it at all and whether it might/may not matter.
- Table 4: would be useful to make the significant t-values Bold text just so they stand out and are easier to see.
- There is no mention of the significant (Table 4) gender and age effects (there needs to be). It would be very useful to have boy-specific and girl-specific subsample analyses to see if results hold similar across sex.
- Very important paper and great measures.
Reviewer 2 Report
There is not a whole lot of interesting conclusions here. What would make this study more thought-provoking would be to show differences between various settings, ie rural, urban schools etc and male female age, grade levels (age differences) if any.
There needs to be more discussion pertaining to activities and mental well-being (different types of well-being).
There is a type in line 15, use male-female consistently, not boy-girl.
